# The One-Dimensional Flow Pressure Loss Correction Model Based on the Particle Flow through Concrete Bend

Guoqiang Gao [1], Lixiang Wei [1], Xuan Zhao [1,2], Minshun Wan [2] and Hongzhou Li [1,*]

[1] School of Mechanical Engineering, Hunan University of Science and Technology, Xiangtan 411201, China; gaoguoqiang2016@163.com (G.G.); weilixiang1999@163.com (L.W.); xuanzhao1998@163.com (X.Z.)
[2] Zoomlion Heavy Industry Science and Technology Development, Changsha 410013, China; 157@zoomlion.com
* Correspondence: hzli@hnust.edu.cn

**Abstract:** The rheological behavior of concrete pumping in the bend section is complex. The existing conversion calculation for pumping pressure loss based on the straight pipe section often fails to meet engineering precision requirements. This paper develops a pressure loss correction model for a concrete pumping bend section based on a new one-dimensional flow model for the straight pipe section simulation of particle flow; the study analyzes the impact of parameters such as elbow shape and pumping flow rate on pressure loss. An equivalent conversion relationship between the bend section and the straight section is established. The comparison with the measured pressure values from the project shows that the relative error between the pressure loss values calculated by the correction model and the actual measurements is 15.8%.

**Keywords:** concrete pumping; bend section; one-dimensional flow model; pressure loss; particle flow

## 1. Introduction

With the emergence of ultra-high, ultra-deep, ultra-large-scale, and other complex projects, concrete pumping technology has become a key technology in modern building construction [1,2]. Improving concrete pumping efficiency, reducing wear and tear on pumping pipes, and decreasing energy consumption are essential for enhancing construction efficiency and lowering costs. Generally, excessively high pumping pressure can lead to concrete segregation, increase wear on aggregate particles and the pumping pipeline, and result in higher energy consumption due to greater pumping power requirements. Reasonable planning of the pumping pipeline layout to effectively reduce pressure loss during concrete conveying is crucial for lowering concrete pumping pressure [3].

The pumping pipe of the pump truck comprises various sections, including straight pipes and bends, which are connected by flanges. The straight pipe is widely used because of its low local resistance. In the pumping straight pipe section, the concrete flow exhibits good continuity and a stable flow state. Under the joint action of pump pressure and pipe wall friction in a non-uniform shear field, the aggregates are pushed by larger shear forces towards regions with smaller shear forces, while the mortar is squeezed through the gaps between the aggregates to the pipe wall. As a result, the concrete in the pump pipe is divided into a stable lubrication layer and a plunger layer [4,5]. Choi et al. used ultrasonic detection to measure the thickness of the lubrication layer at different flow rates. They found that the thickness of the lubrication layer was not significantly affected by the particle size of the coarse aggregate [6,7]. Kwon et al. took the thickness of the lubrication layer into account and derived an equation for calculating both the flow rate and the pressure loss [8]. Kaplan et al. observed that concrete exhibited shear flow at higher flow rates and established a relationship between the pumping pressure and the flow parameters of the lubrication layer [9]. Zhao et al. established the one-dimensional flow model for concrete pumping in straight pipe sections, considering the concrete flow characteristics, and they

used Moody's chart to derive the pressure loss value for concrete pumping [10,11]. The bend pipe can flexibly change the direction of concrete flow to ensure that the concrete is transported to the construction site. Compared to the straight pipe section, the flow of concrete in the bend section is more complex and variable due to interactions such as inertia, wall friction, particle collisions, and particle-avoidance collisions. Numerical simulation techniques, including single-phase flow, discrete element methods, and particle flow simulations, are widely used to model the rheological behavior of concrete. Concrete flow is a very difficult mixture to simulate. In this type of work, there is often a calibration phase that links the simulation with the real behavior of the material [12]. Nerella et al. used the CFD method to develop a simulation model to simulate a slider experiment and used single-phase flow to study the general trend of pressure variation with the discharge rate using this model, which yielded satisfactory results [13]. Wei et al. used the *SST k-ω* turbulence model to predict the flow behavior and pressure loss of pumped concrete in the pipeline [14]. Hao et al. simulated the flow behavior of pumped concrete in a pipe based on the discrete element method, considering the contact parameters between coarse aggregate and cement paste particles [15]. Cao et al. carried out a study on the effect of coarse aggregate volume fraction on concrete pumping pressure based on the discrete element method [16]. Jiang et al. used particle flow simulations to analyze the flow characteristics of concrete during pumping and found that the numerically simulated pressure loss values were close to the experimental results [17]. The standard κ-ε turbulence model is a semi-empirical formulation, which is widely used in engineering applications due to its moderate computational volume and relatively high accuracy of solution [18]. Foreign scholars have carried out relevant research on arrest and wear. De schryver et al. investigated fresh concrete pumping arrest using a CFD modeling approach [19]. Liao et al. studied the pumping wear characteristics of a concrete pipeline based on CFD-DEM coupling [20]. The layout of pumping pipes for pump truck boom pumping affects the rheological behavior of concrete. Jiang et al. used particle flow simulations to analyze the impact of conical and combined conveying pipe model parameters on pressure loss and optimized the pipe arrangement to minimize pressure loss [21].

Concrete pumping flow is influenced by the interaction of factors such as pipe layout, aggregate particles, and pumping speed. In the design and construction of pumping systems, it is crucial to consider specific project requirements, concrete characteristics, conveying distance, and pumping height. As the demands for precise engineering management and high-quality concrete control increase, traditional empirical methods, time-consuming high-precision simulations, and costly coiled-tube experiments are proving inadequate for adapting to industry development trends. In optimizing the pumping pipe layout and determining suitable pumping pressure, the Morinage empirical formula [22] for flow resistance in straight pipe sections is considered to lack precision as it does not account for the influence of aggregate particles. The converted horizontal length method [23] equates the horizontal inclination angle and radius of curvature of the pump pipe, while ignoring the influence of pumping speed on pressure loss. Particle flow numerical simulation [24] fully considers the interactions between aggregate and mortar, offering high calculation accuracy. However, the complexity of the model significantly increases computation time, making it unsuitable for the energy-saving optimization and intelligent control of the boom. Therefore, building on the one-dimensional flow model formula for straight pipe section pressure loss proposed earlier, this paper introduces an engineering correction method for pressure loss in curved pipe sections. This method utilizes high-precision granular flow numerical simulations and considers the effects of curvature radius, horizontal inclination angle, and pumping speed on pressure loss.

In summary, a method for correcting the pressure loss of one-dimensional flow in a concrete pumping bend section based on particle flow is proposed in this paper. The paper is structured as follows: Section 2 introduces the experimental measurement and equivalent calculation methods for concrete pumping pressure loss in engineering construction. Section 3 describes the principles of numerical simulation for granular flow, simulates

the typical flow state of concrete pumping in elbows, and verifies the results through experiments. Section 4 introduces a one-dimensional flow model for horizontal straight pipe sections and analyzes the impact of pumping pipe profile parameters and flow rate on pressure loss. It constructs a correction model for pressure loss in curved pipe sections and verifies the validity of this correction method by comparing it with experimental results. Finally, Section 5 summarizes the main conclusions.

## 2. Concrete Pumping Pressure Loss Engineering Calculation

### 2.1. Experimental Measurements

In the process of concrete pumping, the pumping pipe is pressurized to produce circumferential and axial deformation, and the amount of deformation is directly proportional to the value of the pressure within the concrete. Through the real-time monitoring of the stress and strain of the pumping pipe, this has been an effective means to understand the state of concrete pumping. Based on the generalized Hooke's law, the circumferential stress $\sigma_\theta$ and axial stress $\sigma_Z$ of the pumping pipe wall are:

$$\sigma_\theta = \frac{E}{1 - v^2}(\varepsilon_\theta + v\varepsilon_Z) \tag{1}$$

$$\sigma_Z = \frac{E}{1 - v^2}(\varepsilon_Z + v\varepsilon_\theta) \tag{2}$$

where $E$ is the modulus of elasticity, $\varepsilon_\theta$ is the circumferential strain, $\varepsilon_Z$ is the axial strain, and $v$ is Poisson's ratio.

According to the principle of the mechanics of materials, a circular pipe with wall thickness $h$ is subjected to pressure $p$ inside; $a$ and $b$ are the outer and inner diameters of the pipe, respectively, and the circumferential stress $\sigma_\theta$ at the pipe diameter $r$ is:

$$\sigma_\theta = \frac{pa^2}{b^2 - a^2}\left(1 + \frac{b^2}{4r^2}\right) \tag{3}$$

As concrete is pumped, the pressure value gradually decreases along the conveying direction. A conveying distance of length $l$ is selected, with axial and annular strain gauges arranged at overflow sections 1 and 2, respectively, as shown in Figure 1. The measured values of the axial and annular strains are $\varepsilon_{\theta1}$, $\varepsilon_{Z1}$, $\varepsilon_{\theta2}$, $\varepsilon_{Z2}$, respectively, and the change in annular pressure $\Delta p_n$ in this section is:

$$\Delta p_n = \frac{E(b + a)h[(\varepsilon_{\theta2} - \varepsilon_{\theta1}) + v(\varepsilon_{z2} - \varepsilon_{z1})]}{a^2(1 - v^2)(1 + b^2/4r^2)} \tag{4}$$

According to the Concrete Pumping Construction Technical Regulations (JGJ/T10-2011) [23], the conversion coefficient between circumferential and axial pressure in the pumping pipe is 0.9. The axial pressure loss in the pumping pipe can be determined after conversion.

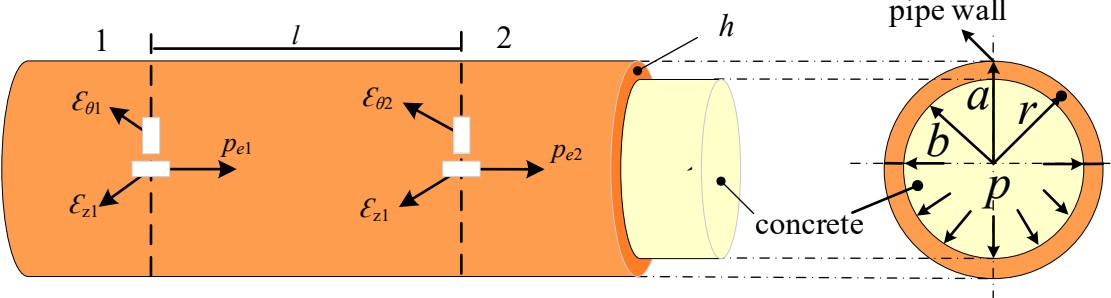

**Figure 1.** Layout of measurement points and stress distribution [10].

### 2.2. Empirical Formula

The value of the pressure loss in the pumping bend section is generally converted to an equivalent value for the horizontal straight pipe section. This conversion often overlooks the pressure loss caused by the flange connections between the bend and the straight pipe, as well as between the straight pipes. For the commonly used pipe diameters of 500 mm and 1000 mm in the market, with a bend pipe tension angle of $\beta$, the pressure loss in the bend section is generally converted using the code method of conversion horizontal length. This method is outlined in the Concrete Pumping Construction Technical Regulations (JGJ/T10-2011) [23]. The details are provided in Table 1. As the tension angle of the bend section increases, the pressure loss also increases. When the tension angle reaches 90°, the pressure loss caused by a single bend is equivalent to that of a 9 m long horizontal straight pipe. In the pumping system, bends are converted to equivalent horizontal straight pipe lengths. The pressure demand for concrete pumping is then determined based on the resistance along the horizontal straight pipe and the elevation difference; that is:

$$p = \Delta p \cdot L + \rho g H \tag{5}$$

where, $p$ is the total pumping pressure, $H$ is the pumping height, $L$ is the equivalent horizontal conveying distance, and $\Delta p$ is the amount of pressure loss per unit length of the horizontal straight pipe.

**Table 1.** Horizontal conversion length of pumped concrete pipes.

| Bend Type | Type of Conversion | Pipe Diameter/Curvature Radius (mm) | Conversion Length (m) |
|-----------|-------------------|-------------------------------------|----------------------|
| Bent Pipes | everyone | 500 | $12\beta/90$ |
| Bent Pipes | everyone | 1000 | $9\beta/90$ |

## 3. Concrete Pumping Particle Flow Simulation

### 3.1. The Principle of Particle Flow Simulation Calculation

During concrete conveying, coarse aggregate particles collide with each other, and the forces involved are influenced by the shape and distribution of the coarse aggregates. This issue is commonly addressed using a combination of finite element and discrete element methods. In concrete conveying simulations, the mortar is treated as a fluid medium, while the aggregate particles are considered to be solid particles flowing within this fluid. The flow behavior of the mortar is calculated using the Eulerian method, while the motion of the solid aggregate particles is analyzed with the Lagrangian method. These two methods are coupled through interaction forces, which include the drag force, buoyancy force, and the pressure generated by the pressure gradient on both sides of the particles. Given the small size of aggregate particles relative to the pumping distance, this paper disregards the pressure generated by the pressure gradient on both sides of the particles. Only the combined effects of drag force and buoyancy are considered. Mortar flow in concrete follows the law of conservation of fluid momentum. The equations of continuity and momentum conservation are used to solve the governing equations, which are expressed as follows:

$$\frac{\partial}{\partial t}(\alpha_f \rho_s) + \nabla \cdot (\alpha_f \rho_s u_s) = 0 \tag{6}$$

$$\frac{\partial}{\partial t}(\alpha_f \rho_s u_s) + \nabla \cdot (\alpha_f \rho_s u_s u_s) = -\alpha_f \nabla P + \nabla \cdot (\alpha_f \mu_s) + \alpha_f \rho_s g - S \tag{7}$$

where, $\alpha_f$ is the volume fraction of mortar to concrete, $\rho_s$ is the density of mortar, $u_s$ is the velocity of mortar, $\nabla P$ is the pressure gradient, $\mu_s$ is the dynamical viscosity of mortar, $S$ is the momentum exchange, and g is the gravitational acceleration.

The motion of concrete aggregate particles includes both translational and rotational components. The state of motion is determined using Newton's second law, described by the following equation:

$$m_i \frac{du_i}{dt} = m_i g + F_{p-g} + \sum_{j=1}^{k_i} \left( F_{cn,ij} + F_{ct,ij} \right) \tag{8}$$

$$I_i \frac{d\omega_i}{dt} = \sum_{j=1}^{k_i} T_{ij} \tag{9}$$

$$F_{p-g} = -\rho_g g V + F_{drag} \tag{10}$$

where, $m_i$ denotes the mass of aggregate particle $i$, $u_i$ denotes the velocity of particle $i$, $\omega_i$ denotes the angular velocity of particle $i$, $F_{p-g}$ denotes the interaction force between the mortar and the particles, $k_i$ denotes the number of particles in contact with particle $i$, $F_{cn,ij}$ and $F_{ct,ij}$ denote the normal and tangential contact forces of particle $i$ and particle $j$, respectively, and $T_{ij}$ denotes the moment induced by the collision of particles $i$ and $j$. $\rho_g$ is the particle density, $V$ is the particle volume, and $F_{drag}$ is the resistance force on the particles. In the concrete conveying process, the Di-Felice traction model is generally chosen to calculate the particle drag force [25]:

$$F_{\text{drag}} = 0.5 C_d \rho_g A \left| v_s - v_g \right| \left( v_s - v_g \right) \alpha_f^{1-\chi} \tag{11}$$

$$\chi = 3.7 - 0.65 \exp \left[ -(1.5 - \lg\text{Re})^2 / 2 \right] \tag{12}$$

$$C_d = \left( 0.63 + 4.8 / \text{Re}^{0.5} \right)^2 \tag{13}$$

where, $C_d$ is the coefficient of traction, $v_s - v_g$ is the relative velocity of the aggregate particles to the fluid, $A$ is the area of the incident surface of the particles into the coarse particles, and $R_e$ is the Reynolds number of concrete.

### 3.2. Particle Flow Numerical Simulation Verification

The bend has a diameter of 125 mm and a vertical (90°) bend with a bend radius of 1000 mm. The density of the pumped C45 concrete mortar is 2100 kg/m³, while the density of the coarse aggregate particles is 2500 kg/m³. The aggregate particle size ranges from 5 to 20 mm and follows a normal distribution. The aggregates have a typical uniform mixture of elongated stripes, flakes, and cones, with these three types of particles frequently selected. The three types of aggregate particles were modeled using a HandySCAN non-contact 3D laser scanner to capture their shapes. Geomagic Design X 2017 software was then used to build the geometric models of these particles. These models were imported into particle flow simulation software, where the aggregate particles were approximated as spherical for simulation purposes, as illustrated in Figure 2. The pumping speeds tested were 0.63 m/s, 0.89 m/s, and 1.13 m/s. The concrete slump was 238 mm, with an extension degree of 451 mm. Aggregate particle bond parameters were calibrated through virtual tests of the concrete slump, as detailed in Table 2.

**Table 2.** Coupling simulation parameter setting.

| Parameter | Value |
| --- | --- |
| Particle Poisson's ratio | 0.35 |
| Pipe Poisson's ratio | 0.30 |
| Particle–particle static friction coefficient | 0.30 |
| Particle–particle rolling friction coefficient | 0.02 |
| Particle–wall static friction coefficient | 0.20 |
| Particle–wall rolling friction coefficient | 0.03 |
| Particle–wall coefficient of restitution | 0.60 |
| Particle–particle coefficient of restitution | 0.70 |

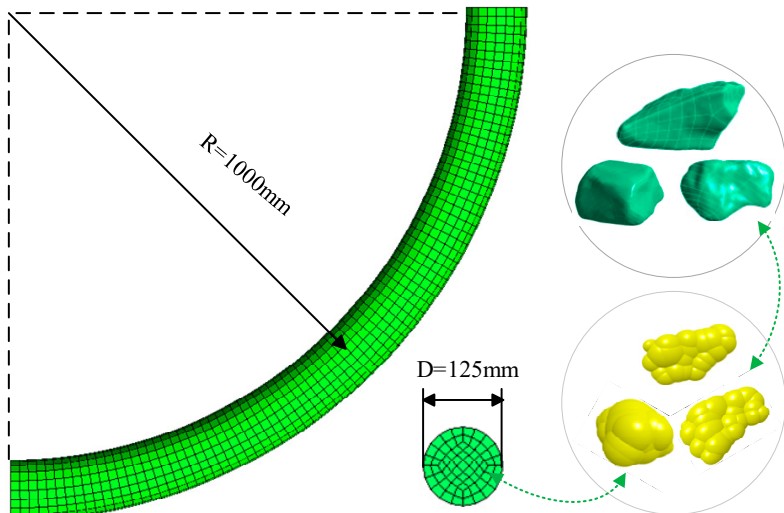

**Figure 2.** Pumping bend model and particle model.

In simulating concrete flow within the pumping pipe, a hexahedral mesh is used; the influence of the number of grids and the sparse distribution of grids on the numerical simulation is considered when performing numerical simulation. Considering the computational accuracy and efficiency, this paper adopts 7270 grid elements and 6460 nodes after checking for mesh independence to carry out the simulation analysis of CFD. The inlet is set with a velocity inlet boundary condition, and the wall is modeled as a no-slip boundary. The outlet section of the bend is atmosphere. The standard $\kappa$-$\varepsilon$ turbulence model is employed. The fluid simulation time step is $10^{-4}$ s, with a convergence accuracy of $10^{-3}$. The generation rate of coarse aggregate is determined by the pumping speed and the aggregate particle content. The time step for the discrete element calculation of aggregate movement is $10^{-6}$ s, and the total simulation time for the coupling of mortar and particles is 2 s. The numerical simulation is performed using an 11th Gen Intel(R) Core(TM) i7-11700 @ 2.50 GHz processor, and the simulation results are presented in Figure 3. The average pressure difference between the inlet and outlet sections represents the pressure loss in the bend. This simulated pressure loss was compared with the experimental data, as shown in Table 3. The relative errors between the simulated and experimental pressure loss values are less than 10%. The feasibility of using particle flow simulations to model concrete pumping behavior is demonstrated.

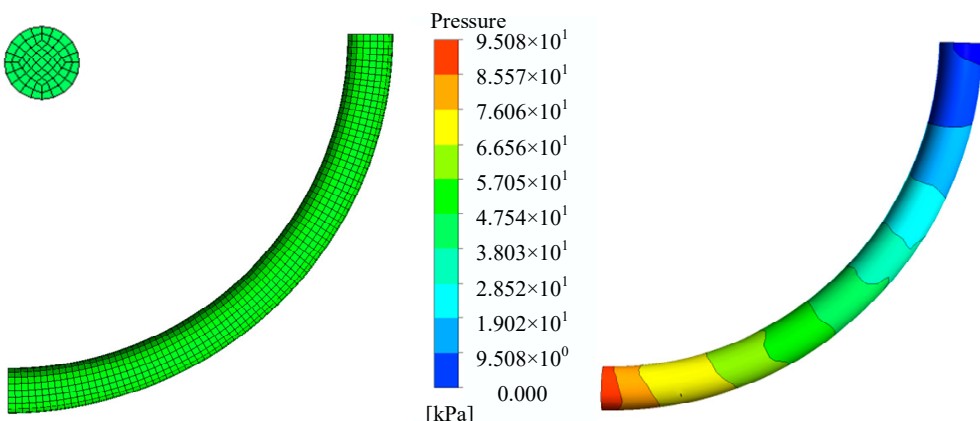

**Figure 3.** The pressure distribution at 1.45 s with a pump velocity of 1.13 m/s.

**Table 3.** Comparison of experimental data and simulation results.

| Average Flow Velocity | Experimental Data Value | Simulated Calculation of Pressure Loss Value | Relative Error |
|---|---|---|---|
| 0.63 m/s | 90.1 KPa/m | 83.6 KPa/m | −6.59% |
| 0.89 m/s | 96.8 KPa/m | 89.6 KPa/m | −7.43% |
| 1.13 m/s | 102.3 KPa/m | 93.8 KPa/m | −8.28% |

## 4. Pressure Loss Correction Model for Bend Section

### 4.1. One-Dimensional Flow Model for Horizontal Straight Pipe Section

In the pumping pipe, the concrete is divided into a plunger layer and a lubrication layer. In the plunger layer, aggregate particles and mortar move together as a cohesive unit. This results in minimal friction between the particles and the mortar, which corresponds to a lower pressure loss. In the lubrication layer, the mortar flows in layers, experiencing friction between the mortar and the pumping pipe wall, friction due to the relative slip between different mortar layers, and friction from the relative movement between the lubrication layer and the plunger layer. This friction results in significant energy consumption and corresponds to considerable pressure loss during the pumping process. The factors affecting pumping pressure loss include pipe wall roughness, lubrication layer thickness, aggregate particle shape and size, viscosity of the lubrication layer, Reynolds number, and concrete pumping speed. The one-dimensional flow model assumes that the lubrication layer and the critical point of the plunger layer are filled with coarse aggregate particles. Coarse aggregate particles of different shapes are approximated as spherical particles with equivalent sizes, as illustrated in Figure 4, where $v$ represents the average concrete velocity, $l$ is the distance between the selected control body overflow sections 1 and 2, $z_1$ is the height of the axis of overflow section 1 from the reference datum, $z_2$ is the height of the axis of overflow section 2 from the reference datum, $p_1$ is the average pressure at overflow section 1, $p_2$ is the average pressure at overflow section 2, $d$ is the inner diameter of the pumping pipeline, $\delta$ is the thickness of the lubrication layer, $K$ is the equivalent particle size of the coarse aggregate particles, and $\tau_0$ is the viscous force between the lubrication layer and the plunger layer.

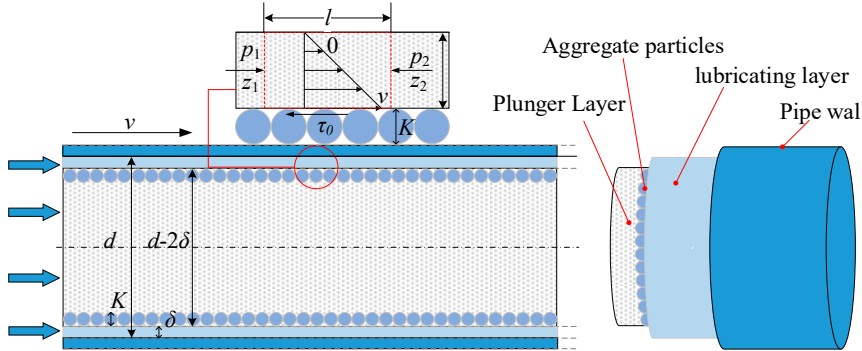

**Figure 4.** Schematic diagram of one-dimensional flow model for concrete pumping [10].

To calculate the pressure loss along the pumping pipeline, the Darcy formula is used, with the along-range loss coefficient determined by Moody's chart. This coefficient is primarily dependent on the Reynolds number and relative roughness during concrete conveying. The pressure difference $\Delta p$ between the two cross-sectional streams of concrete is calculated as follows:

$$\Delta p_m = 0.0055 \left[ 1 + \left( 20000 K_d + \frac{10^6}{\text{Re}} \right)^{\frac{1}{3}} \right] \frac{l \rho_s v^2}{2 d_e} \tag{14}$$

where, $d_e$ is the equivalent diameter of the lubrication layer, $K_d$ is the relative roughness of the lubrication layer in contact with the plunger layer, and Re is the Reynolds number of the mortar in the lubrication layer. The lubrication layer is the annular region formed by the pumping pipe wall and the plunger layer, and its equivalent diameter $d_e$ is:

$$d_e = 2\delta \tag{15}$$

Relative roughness $K_d$ is defined as the ratio of absolute roughness $K$ to the equivalent diameter $d_e$. According to the one-dimensional flow model assumption, the area where the lubrication layer contacts the plunger layer consists of coarse aggregate particles with uniform particle size. Thus, the absolute roughness is considered to be the average particle size of the aggregate particles. Aggregate particles have complex shapes and different sizes. The HandySCAN non-contact 3D laser scanner and reverse modeling software Geomagic Design X are used to geometrically reconstruct the aggregate particles. The total surface area $S_a$ of the selected aggregate particles is calculated using the geometric body surface area statistics function of the 3D mapping software(Geomagic Design X 2017 software), as shown in Figure 5. To equate the hydraulic radius of the non-spherical particles to that of the spherical particles, the equivalent diameter $K$ of the non-spherical particles is calculated as follows:

$$K = {6V_a}\Big/{S_a} \tag{16}$$

where $Va$ is the total volume of the selected aggregate particles.

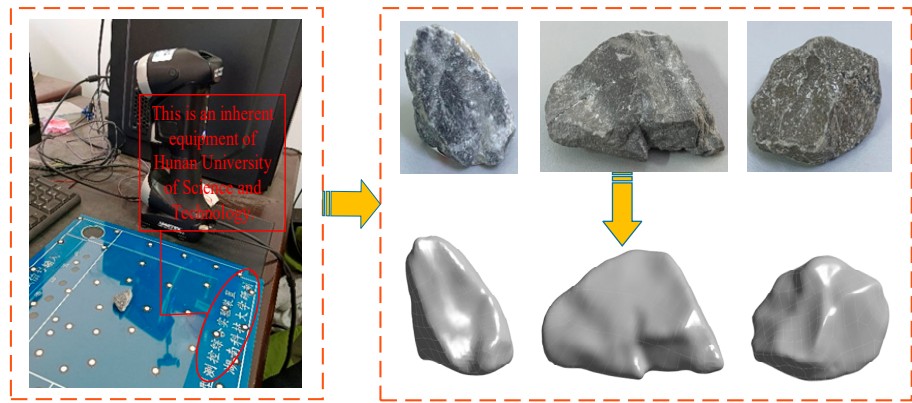

**Figure 5.** Geometric reconstruction of coarse aggregate [10].

### 4.2. Sensitivity Analysis of Parameters in Bend Section

Horizontal inclination affects the configuration of the pumping pipe on the pump truck. An excessively large or small radius of curvature can increase the risk of pipe clogging. Additionally, concrete conveying resistance varies with changes in both the horizontal inclination and the radius of curvature of the pumping pipe. Concrete conveying resistance varies with different pumping flow rates. Therefore, it is important to analyze how the horizontal inclination angle of the pumping pipe, the radius of curvature, and the pumping flow rate affect concrete pressure loss. Additionally, since the curved pipe tension angle is typically 90° in the current market, this paper does not include a sensitivity analysis of the curved pipe tension angle.

According to the particle flow numerical simulation process, with a concrete pumping pipeline diameter of 125 mm, the combined effects of centrifugal force, particle collisions, and collisions with the pipe wall create blocking effects at the elbow. To better simulate the concrete flow, an adjustment section is added: a 500 mm straight pipe section is placed before the elbow, and a 400 mm straight pipe section is added after the elbow. The concrete model and aggregate particle shape remain consistent with the validation model, with a radius of curvature of 195 mm and a pumping speed of 0.905 m/s. The inlet axis is

parallel to the horizontal plane, and the outlet axis is at an angle to the horizontal plane. The pumping pipes are arranged at angles of 90°, 45°, 0°, −45°, and −90° relative to the horizontal direction. The bend model is meshed using hexahedral elements, totaling 5470 elements and 5092 nodes. The simulation duration is 2 s, with a fluid simulation time step of 0.0001 s and a particle simulation time step of 0.000001 s. The interaction forces between the fluid and the particles are calculated using a coupled drag force model. After the numerical simulation of particle flow, the pressure distribution in the pumping pipeline is as illustrated in Figure 6. The figure shows that the pressure at the pipeline inlet increases with the horizontal inclination angle. When the inclination angle is perpendicular to the ground, the pressure loss during concrete pumping is highest. This is because, in addition to overcoming resistance along the pipeline, the concrete must also overcome the flow resistance generated by its own weight. In other words, as the horizontal inclination of the pipeline increases, the concrete must overcome not only the increased resistance to flow caused by its own weight but also greater flow resistance within the pipe. This leads to higher energy loss.

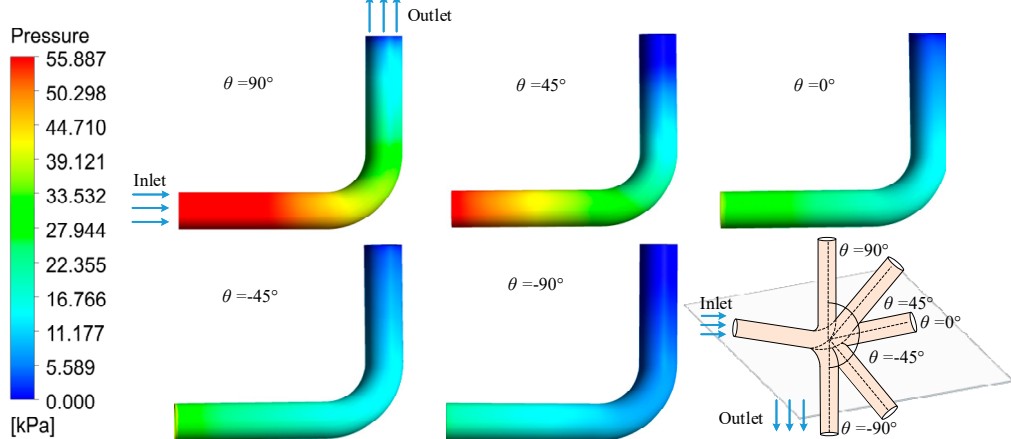

**Figure 6.** Bend section of different horizontal inclination pressure distribution cloud map.

For a concrete pumping pipeline with a diameter of 125 mm, a horizontal arrangement of the bend, and a flow rate of 0.905 m/s. The simulation analysis for C45 concrete is conducted using curved pipes with radii of curvature of 195 mm, 235 mm, 275 mm, 315 mm, and 355 mm. The pressure distribution in the pipe is illustrated in Figure 7. The figure shows that as the radius of curvature increases, the average pressure at the inlet section of the bend also increases. This is because a larger radius of curvature extends the pumping time and path, which raises the energy required to overcome resistance along the pipeline. Consequently, the increased pressure loss due to the larger radius of curvature results in a higher average pressure at the inlet.

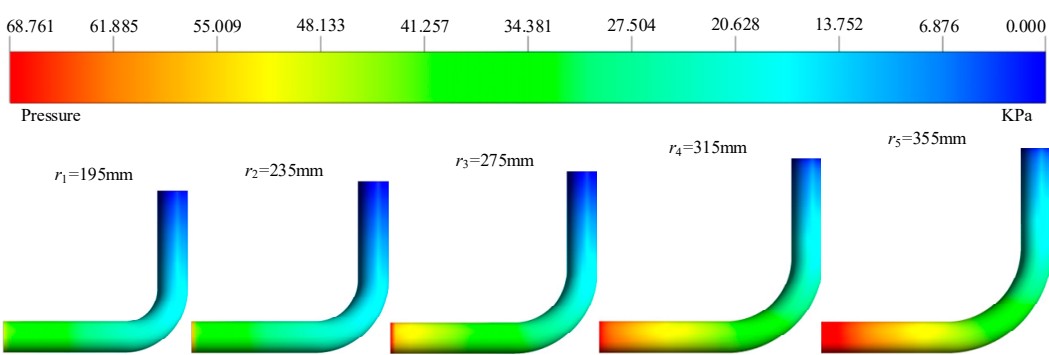

**Figure 7.** Pressure distribution of different radii of curvature in the bend section.

A bend with a curvature radius of 195 mm is used, and the pumping flow rates are 40 m³/h, 55 m³/h, 70 m³/h, 100 m³/h, and 120 m³/h. The elbow is positioned at a 90° inclination to the horizontal direction. Numerical simulation analysis of the particle flow is performed using the standard $\kappa$-$\varepsilon$ turbulence model. The calculation results are shown in Figure 8. The figure shows that as the pumping flow rate increases, the average pressure at the inlet section also rises. This is because a higher flow rate significantly impacts the rheological properties of the concrete. This affects the velocity distribution within the pipeline and, consequently, the pressure distribution.

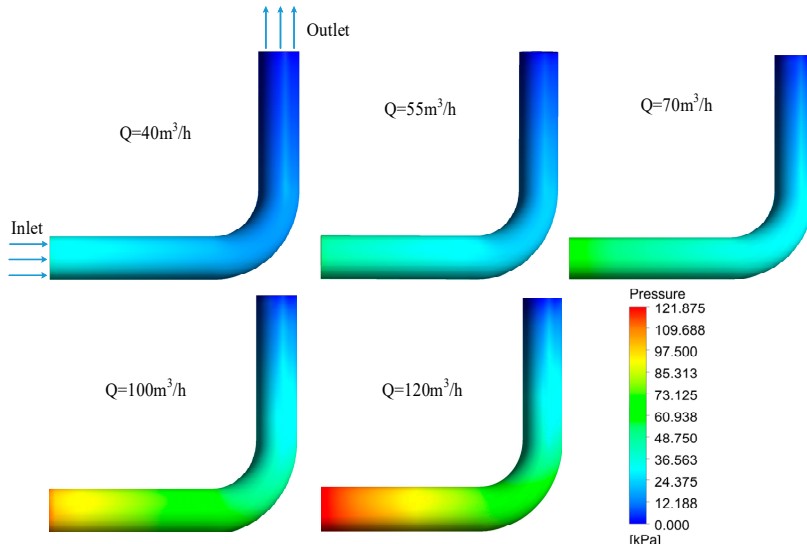

**Figure 8.** Pressure distribution cloud at different pump flow rates.

*4.3. Correction Model*

This paper uses the one-dimensional flow model of a horizontal straight pipe section for concrete pumping as a reference. It considers the effects of the bend radius, horizontal inclination angle, and pumping flow rate on the pressure loss in the bend. A correction model for the pressure loss of the bend is then established. The pressure loss value $\Delta p_w$ of the bend is given by:

$$\Delta p_w = \lambda \Delta p_m = \lambda 0.0055 \left[ 1 + \left( 20000 K_d + \frac{10^6}{\text{Re}} \right)^{\frac{1}{3}} \right] \frac{l \rho_s v^2}{2 d_e} \tag{17}$$

where, $\lambda$ is the correction factor.

The bend radius, horizontal inclination angle, and flow rate directly affect the pressure loss in concrete pumping. To further explore the impact of these parameters, a three-factor, five-level orthogonal experimental design is used. The inclination angles of the pumping pipeline relative to the horizontal are 90°, 45°, 0°, −45°, and −90°. The concrete pumping speeds are 40 m³/h, 55 m³/h, 70 m³/h, 100 m³/h, and 120 m³/h. The selected curved pipe radii of curvature are 195 mm, 235 mm, 275 mm, 315 mm, and 355 mm. To enhance the simulation of concrete flow, a straight pipe section of 500 mm is added before the elbow, and a straight pipe section of 400 mm is added after the elbow. The aggregate particle shape and physical parameters are consistent with those in the validation model. A total of 125 particle flow simulation analyses are conducted, with the results detailed in Table 4.

**Table 4.** Orthogonal experimental particle flow simulation analysis results (MPa/m).

| Horizontal Inclination Angle | Pumping Flow Rate (m³/h) | Radii of Curvature (mm) | | | | |
|---|---|---|---|---|---|---|
| | | 195 | 235 | 275 | 315 | 355 |
| 0° | 40 | 0.1884 | 0.1987 | 0.1776 | 0.1619 | 0.1499 |
| | 55 | 0.2632 | 0.2763 | 0.2408 | 0.2259 | 0.2075 |
| | 70 | 0.3305 | 0.3616 | 0.3202 | 0.2874 | 0.2638 |
| | 100 | 0.4895 | 0.5287 | 0.4673 | 0.4268 | 0.3912 |
| | 120 | 0.6079 | 0.6555 | 0.5757 | 0.5158 | 0.4750 |
| 45° | 40 | 0.2376 | 0.2279 | 0.2086 | 0.1938 | 0.1801 |
| | 55 | 0.3098 | 0.3238 | 0.2876 | 0.2649 | 0.2420 |
| | 70 | 0.3872 | 0.4058 | 0.3552 | 0.3226 | 0.3037 |
| | 100 | 0.5425 | 0.5749 | 0.5124 | 0.4613 | 0.4232 |
| | 120 | 0.6572 | 0.7031 | 0.6181 | 0.5555 | 0.5121 |
| 90° | 40 | 0.2622 | 0.2448 | 0.2243 | 0.2083 | 0.1937 |
| | 55 | 0.3368 | 0.3432 | 0.3028 | 0.2794 | 0.2503 |
| | 70 | 0.4135 | 0.4206 | 0.3775 | 0.3432 | 0.3221 |
| | 100 | 0.5641 | 0.5966 | 0.5258 | 0.4726 | 0.4334 |
| | 120 | 0.6789 | 0.7208 | 0.6327 | 0.5742 | 0.5247 |
| −45° | 40 | 0.1430 | 0.1522 | 0.1368 | 0.1241 | 0.1140 |
| | 55 | 0.2205 | 0.2373 | 0.2109 | 0.1959 | 0.1783 |
| | 70 | 0.2920 | 0.3220 | 0.2838 | 0.2557 | 0.2456 |
| | 100 | 0.4465 | 0.4902 | 0.4322 | 0.3892 | 0.3589 |
| | 120 | 0.5627 | 0.6216 | 0.5411 | 0.4850 | 0.4423 |
| −90° | 40 | 0.1176 | 0.1261 | 0.1187 | 0.1099 | 0.0993 |
| | 55 | 0.2023 | 0.2176 | 0.1913 | 0.1733 | 0.1574 |
| | 70 | 0.2810 | 0.3048 | 0.2667 | 0.2427 | 0.2157 |
| | 100 | 0.4331 | 0.4778 | 0.4155 | 0.3734 | 0.3414 |
| | 120 | 0.5388 | 0.5978 | 0.5249 | 0.4721 | 0.4324 |

The relationship between the horizontal inclination of the bend and the pressure loss value is shown in Figure 9. For a flow rate of 40 m³/h and a bend radius of 195 mm, the pressure loss increases with a larger horizontal inclination. The correlation between the horizontal inclination angle and the pressure loss is linear, with a goodness-of-fit ($R^2$) value of 0.9865. When the bend radius is 235 mm, 275 mm, 315 mm, or 355 mm, the pressure loss value relative to the horizontal inclination angle remains essentially consistent, following a linear distribution. The $R^2$ values exceed 0.97, indicating a good fit. At the same time, a larger curvature radius of the pumping pipeline results in lower pressure loss during concrete transportation. However, the spacing between each curve changes, indicating an interactive relationship between the pipeline curvature radius and the horizontal inclination angle.

The relationship between the bend radius and concrete pumping pressure loss is illustrated in Figure 10. As the bend radius increases, the pumping pressure loss decreases when the horizontal inclination of the bend is 90° and the flow rate is 40 m³/h. At different flow rates, the relationship between pressure loss and bend radius exhibits similar characteristics. At the same time, as the flow rate increases, the corresponding pressure loss value also increases. Notably, the spacing between the curves changes, indicating a significant interaction between the bend radius and the flow rate.

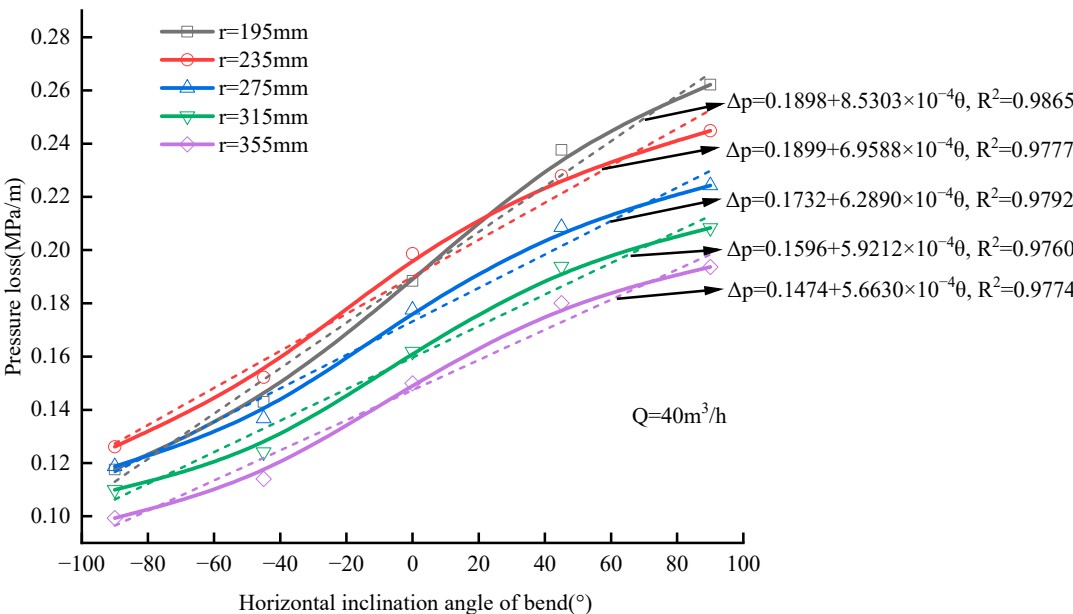

**Figure 9.** The curve of the distribution of pressure loss values with the horizontal inclination angle of bend.

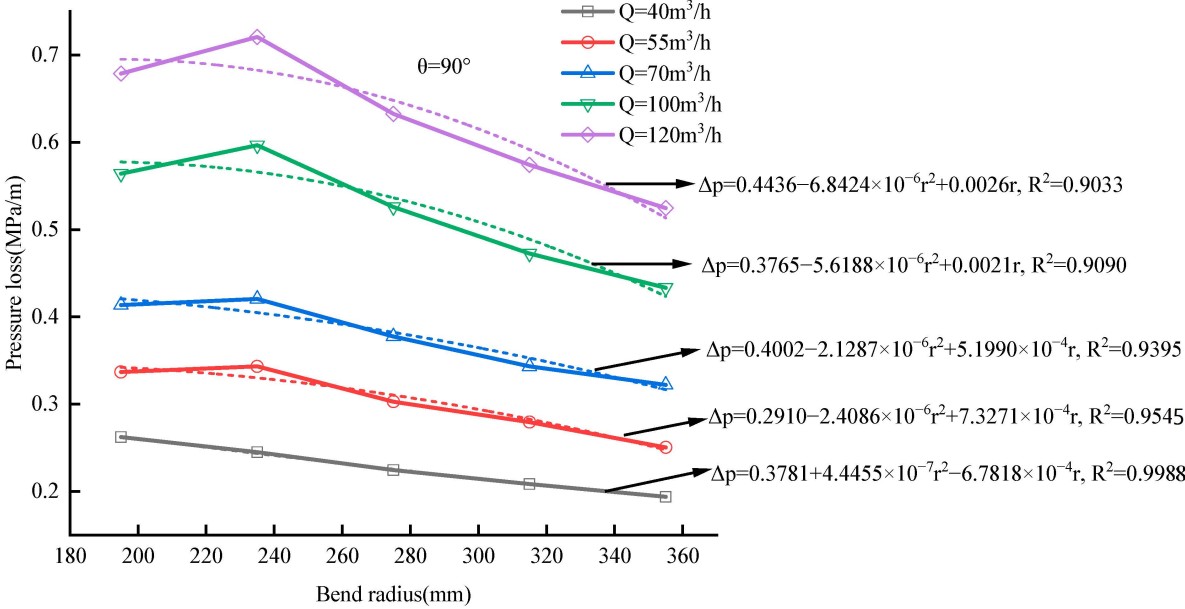

**Figure 10.** The curve of the distribution of pressure loss in concrete pumping with bend radius.

Figure 11 primarily illustrates the pressure loss associated with varying flow rates at different horizontal inclination angles. The figure shows that the flow rate and the concrete pumping pressure loss exhibit a clear linear relationship, with the $R^2$ exceeding 0.99. As the pumping flow rate increases, the pressure loss also rises. This linear trend remains consistent even when the layout of the pumping pipeline changes. Additionally, as the horizontal inclination angle increases, the corresponding pressure loss value also increases accordingly.

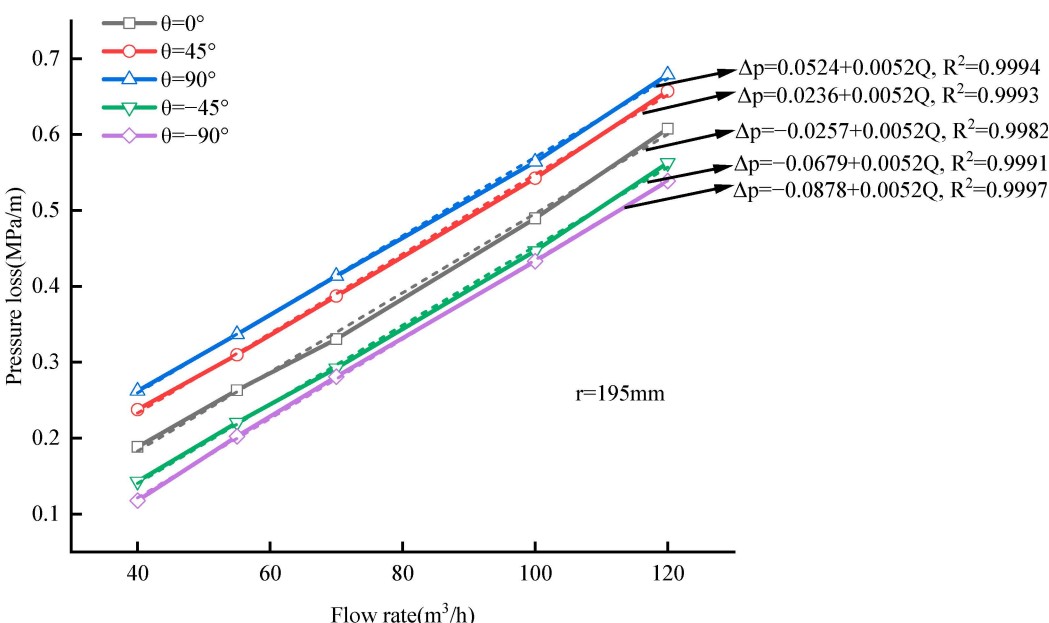

**Figure 11.** The curve of the distribution of pressure loss in concrete pumping with the flow rate.

Therefore, by considering the combined effects of horizontal inclination, radius of curvature, and pumping flow rate on the pumping pressure loss, along with the interaction between these factors, the expression for the pumping bend correction factor is:

$$\lambda = c_1 r^2 \theta + c_2 r \theta + c_3 v \theta + c_4 r v + c_5 r + c_6 \theta + c_7 v + c_8 \tag{18}$$

The results of the pressure loss conversion in Table 4 are fitted using the least squares method. The fitted results are shown below:

$$
\begin{aligned}
\lambda_1 = &-2.0663 \times 10^{-8} r^2 \theta + 1.2749 \times 10^{-5} r \theta - 0.00157 v \theta + 0.0021 r v \\
&+0.0025 r + 0.00807 \theta + 0.7545 v + 0.3001
\end{aligned}
\tag{19}
$$

The fitting results reflect the combined effect of pumping velocity, horizontal inclination, and the radius of curvature of the bend on the pumping pressure loss in the elbow section of the pipeline.

### 4.4. Verification of Pressure Loss in Bend Section

To validate the effectiveness of the equivalent pressure loss model for the elbow section, this study uses the field parameters from [26]. A high-rise residential building has 25 floors, with a total height of 75 m. The first vertical bend of concrete pumping is selected for strain monitoring, and the project adopts the Zoomlion ZL140THBE-10022R truck-mounted pump (Zoomlion Heavy Industry Science and Technology Co., Ltd., Changsha, China). The project adopts the ZLJ140THBE-10022R truck-mounted pump; the maximum theoretical conveying capacity is 100 m$^3$/h, the concrete cylinder diameter is 200 mm, and the cylinder stroke is 1650 mm. The specific experimental results refer to those in [26]. The geometric model matches the one described in Section 3.2 of this paper. The elbow has a horizontal inclination angle of 90°, an opening angle of 90°, an inner diameter of 125 mm, and a curvature radius of 1000 mm. The pump truck operates at a pressure of 20 MPa, with the total height of the residential building under construction being 75 m. The concrete has a slump of 238 mm and an expansion of 415 mm.

According to the converted horizontal length method outlined in the Technical Regulations for Concrete Pumping Construction (JGJ/T10-2011), the pressure loss of a curved pipe with a tension angle of 90° is equivalent to that of a 9 m long horizontal straight pipe.

Using Morinage's empirical formula, the pressure loss $\Delta p_c$ per unit length of a horizontal straight pipe is:

$$\Delta p_c = \frac{2}{R}[k_1 + k_2(1 + t_2/t_1)v]\alpha \tag{20}$$

where $\alpha$ is the ratio of radial pressure and axial pressure, usually taken as 0.90; $R$ is the inner diameter of the pipe; $k_1$ is the resistance coefficient produced by the concrete sticking to the pipe; $k_2$ is the resistance coefficient produced by the uneven speed of concrete in the pipe section; $t_1$ is the pumping piston reciprocating motion to promote the time of concrete; $t_2$ is the time of the distribution valve conversion; $t_2/t_1$ is generally taken as 0.3. $k_1$ and $k_2$ are related to the concrete mixing ratio, pipe wall smoothness, and other factors related to the engineering of commonly used concrete; slump S is used for calculation.

The type of concrete conveyed by the pump truck is C45. The kinematic viscosity coefficient of the concrete mortar is 10.6 m$^2$/s, and the density of the aggregate particles is 2182 kg/m$^3$. The thickness of the lubrication layer in the pumped straight pipe section is 2.1 mm, the equivalent diameter is 4.2 mm, and the relative roughness between the lubrication layer and the plunger layer is 3.76 mm. For a pumping flow rate of 0.63 m/s, the pressure loss per unit length, calculated using Morinage's empirical formula, is 0.0056 MPa. When the pumping flow rates are 0.77 m/s and 0.89 m/s, Morinage's empirical formula calculates the pressure losses per unit length as 0.0065 MPa and 0.0072 MPa, respectively.

This paper uses a one-dimensional flow model to calculate the pressure loss in the elbow section. For comparison, the pressure loss in the straight pipe section is also calculated using this model. The results show that, for pumping flow rates of 0.63 m/s, 0.77 m/s, and 0.89 m/s, the pressure losses per unit length are 0.0224 MPa, 0.0214 MPa, and 0.0219 MPa, respectively. These values are higher than those calculated using Morinage's empirical formula. The values of the pressure loss in bends calculated using Morinage's empirical formula and the pressure loss in straight pipe sections determined by the one-dimensional flow model are referenced. The results, using the converted horizontal length method and the modified one-dimensional flow model formula, are presented in Table 5. The experimental test values are sourced from [26]. The results indicate that the pressure loss values for the bends, calculated using the combination of Morinage's empirical formula and the converted horizontal length method, are consistently lower than the experimental test values. The relative error compared to the experimental values exceeds 30%, with the error reaching 43.95% at a flow velocity of 0.63 m/s. The pressure loss values calculated using the combination of the one-dimensional flow model and the converted horizontal length method are consistently higher than the experimental test values, approximately twice the magnitude of the experimental results. The pressure loss value calculated using the correction formula in this paper is closer to the experimental test values, with the relative error controlled within 20%. Notably, the relative error is as low as 11.16% when the pumping speed is 0.77 m/s. There are two main reasons for this result. Firstly, the Morinage empirical formula significantly underestimates the pressure loss in the straight pipe section compared to the actual measurements. According to [10], the Morinage formula also shows a discrepancy in typical straight pipes, with a relative error exceeding 50% when compared to the experimental results. Secondly, the horizontal length conversion method only accounts for the bend's tension angle and radius of curvature. The impact of horizontal inclination and pumping speed on pressure loss is neglected. This omission affects the accuracy of the calculations. This paper's particle flow numerical simulations reveal that the pressure loss in the bend section is influenced not only by the radius of curvature but also by pumping speed, horizontal inclination, and other factors. This paper builds upon the straight pipe section pressure loss calculated using a one-dimensional flow model. By analyzing the influence of pumping speed, radius of curvature, and horizontal inclination on the pressure loss in the bend section, the formula for pressure loss in the bend section has been revised. The results indicate that the amended formula meets the engineering accuracy requirements, thereby validating the effectiveness of the method.

**Table 5.** Equivalent conversion values for bends with measured parameters.

| Pumping Flow Rate (m/s) | Morinage Empirical Formula + the Converted Horizontal Length Method | One-Dimensional Flow Model + the Converted Horizontal Length Method | One-Dimensional Flow Correction Model | Experimental Measurement Values [26] |
|---|---|---|---|---|
| 0.63 | 0.0505 | 0.2016 | 0.0995 | 0.0901 |
| 0.77 | 0.0581 | 0.1926 | 0.1042 | 0.0942 |
| 0.89 | 0.0647 | 0.1971 | 0.1150 | 0.0968 |

## 5. Conclusions

In this paper, by applying particle flow simulation principles and fluid dynamics theory and integrating a new one-dimensional flow model for concrete pumping straight pipe sections, an engineering calculation model for bend pressure loss has been developed. The accuracy of the original engineering calculations is enhanced by this model. The main conclusions are:

(1) The relative error between the concrete pumping pressure loss values calculated by particle flow simulation and the actual measured data is less than 10%. The accuracy of the numerical simulation method is confirmed by this result.

(2) The radius of curvature, horizontal inclination, and pumping flow rate significantly affect the pressure loss in the pumping bend section.

(3) The error between the pressure loss values calculated using the bend pressure loss correction model and the experimental data is less than 20%. This accuracy enhancement of the converted horizontal length method is achieved.

**Author Contributions:** Conceptualization, G.G.; methodology, G.G.; software, L.W.; validation, L.W.; formal analysis, X.Z.; investigation, X.Z.; resources, M.W.; data curation, X.Z.; writing—original draft preparation, G.G.; writing—review and editing, H.L.; visualization, L.W.; supervision, H.L.; project administration, M.W. and H.L.; funding acquisition, G.G. All authors have read and agreed to the published version of the manuscript.

**Funding:** This research was supported by the National Natural Science Foundation of the People's Republic of China (grant number 51805163).

**Institutional Review Board Statement:** Not applicable.

**Informed Consent Statement:** Not applicable.

**Data Availability Statement:** The data presented in this study are available on request from the corresponding author due to privacy.

**Conflicts of Interest:** The authors declare no conflict of interest.

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
