# Peer review of "The One-Dimensional Flow Pressure Loss Correction Model Based on the Particle Flow through Concrete Bend"

_applsci, doi:10.3390/app14198824_

Round 1

Reviewer 1 Report

Comments and Suggestions for Authors

Here are some questions, doubts and major remarks from my side:

1.       The title “The One -Dimensional Flow Pressure Loss Correction Model Based on the Concrete Pumping Bend of Particle Flow” is not understandable. Maybe: “The One -Dimensional Flow Pressure Loss Correction Model Based on the Particle Flow through Concrete Pumping Bend

2.       What is an elbow and what is a bend? Switching from elbow to bend makes matters unclear. In my opinion, you can use bend, it is more comprehensive.

3.       What is Modi¢s formula? More likely it is Moody¢s Chart. You cited as reference paper „One-Dimensional Modeling of the Pressure Loss in Concrete Pumping and Experimental Verification“ by Xuan Zhao, Guoqiang Gao, Minshun Wan and Juchuan Dai, but the source is not mentioned there.

4.       There are figures in this paper that are the same or very similar to those in the paper „One-Dimensional Modeling of the Pressure Loss in Concrete Pumping and Experimental Verification“ by Xuan Zhao, Guoqiang Gao, Minshun Wan and Juchuan Dai, and it is not mentioned in the text.

5.       To make this subject clearer it would be nice to provide a sketch of the elbow in which you can sign the horizontal inclination angle and opening angle.

6.       Excessive use of the term „pumping“, such as “concrete pumping”, “pumping bend”, “pumping pipeline”,… I think that pumping can be omitted in a majority of the text. It is a known fact that the concrete flows through the pipeline thanks to pumping.

Here are some remarks that should be considered:

Line 11: "The pressure loss values calculated by converting to the straight section"

Line 37: what do you mean under “small regions”? Do you mean “regions with smaller shear forces”?

Line 46: "an one-dimensional"

Line 47: I have never met Modi` formula so far. Maybe the authors are referring to Moody`s chart to determine coefficient of friction and then use Darcy-Weisbach equation to calculate pressure drop?

Line 110: h is subjected to pressure p inside (italic!)

Line 111: r is

Line 123: Fig. 1 4 is almost the same as the figure in the paper “One-Dimensional Modeling of the Pressure Loss in Concrete Pumping and Experimental Verification“ by Xuan Zhao, Guoqiang Gao, Minshun Wan and Juchuan Dai

Line 131: "horizontal length method. This method"

Line 183: "Rep is the Reynolds number"

Line 186: The concrete pumping pipe has a diameter of 125 mm and a vertical (90°) bend with a

Line 202: You should write down that your outlet section of the bend is atmosphere!

Line 234: Where, v represents the average concrete velocity! It is not flow rate!

Line 242: Figure 4 is almost the same as the figure in the paper “One-Dimensional Modeling of the Pressure Loss in Concrete Pumping and Experimental Verification“ by Xuan Zhao, Guoqiang Gao, Minshun Wan and Juchuan Dai

Line 245: “with the along-range loss coefficient determined by Modi's formula. Where is the source of this formula?

along-range loss coefficient determined by Modi's formula” It is a coefficient of friction determined from Moody's chart.

Line 255: the ratio of absolute roughness K to the equivalent

Line 258: Thus, the absolute roughness is considered

Line 264: You are using notation K for the equivalent diameter and for absolute roughness, it should be changed

Line 269: Fig. 54 is the same as the figure in the paper “One-Dimensional Modeling of the Pressure Loss in Concrete Pumping and Experimental Verification“ by Xuan Zhao, Guoqiang Gao, Minshun Wan and Juchuan Dai

Line 299: excessive use of „pumping“; „flow resistance within the pumping pipe“ 

Line 304-307: excessive use of „pumping“ in “the pumping bend“, as well as in „pumping speed of 0.905 m/s“ and „pumping pipe“. Pumping is surplus here, and pumping speed mostly refers to flow rate, not to the velocity. Bend and elbow. You can define both, but in my opinion it is better to use one term, more comprehensive: bend

Line 308: What is “the pump inlet section of the elbow“? Isn't is sufficient to say the inlet section of the elbow (i.e. bend)“?

So, you have assumed that the outlet section of the bend is with atmospheric pressure (0 Pa)? Actualy, you are calculating pressure drop through bend.

Line 314 and 327: pimping elbow i.e. bend

Line 326: “pipe radius of curvature“, better: „bend radius“

Line 327 and 328: „pumping elbow“ better: „bend“; „elbow pipe“ better: „bend“

Line 328 and 329: elbow pipe

Line 332: “The radius of curvature of the elbow bend radius, horizontal inclination, and pumping flow rate

Line 344-346: The relationship between the horizontal inclination angle of the bend pipe and the pressure loss value is shown in Fig. 9. For a pumping flow rate of 40 m³/h and a pipe  curvature bend radius of 195 mm, the pressure loss increases with a larger horizontal inclination angle.

Line 348: “When the curvature radius of the pumping pipeline is“ better: „When the bend radius is

Line 358 and 359 and 362 and 365: The relationship between the curvature bend radius of the elbow and concrete pumping

Line 366 and 368: Bend radius of curvature(mm)

Line 385:  It is velocity and not flow rate!

Line 405: „t1 is the pumping piston reciprocating motion to promote the time of concrete“ I do not understand this. t1 is time, isn¢t it?

Line 457: concrete pumping elbow bend

Comments on the Quality of English Language

In my opinion, the language in which this paper is written is at a satisfactory level, although there are some places that I have indicated, where I recommend some modifications. Everything is contained in the attached document.

Reviewer 2 Report

Comments and Suggestions for Authors

Line 195 as illustrated in Fig. 2. should be as illustrated in Fig. 2. (there should be a dot after fig.)

Fig. 3 a) is the number grid too sparse and should it be thickened at the perimeter in the wall layer due to linear and local pressure losses?

Line 293 end of sentence and start of sentence from fig.5.. (so should not be started sentence)

Line 307 end of sentence and start of sentence from fig.7. (so should not be started sentence)

Line 318 end of sentence and start of sentence from fig.8. (so should not be started sentence) moreover there are editorial errors.

Line 370 angles. Fig.11 shown should be angles. Fig. 11 shown

Figs.10 and 11 Did they correctly use the coefficient of determination R^2 in the regression function? Did the authors not care to linearise and specify the linear correlation coefficient in the regression function?

Line 384 no formula number, being a substitution for formula 17 from line 382.

Line 453 very illegible is the header describing the table and the actual table on the next page I believe there is little conclusion and reference to the actual application in the summary What do these considerations and results actually yield References The literature is up to date, however, is this type of recent research mostly or only conducted by researchers from Chinese universities?

I believe there should be more in-depth literature analysis in the World

Reviewer 3 Report

Comments and Suggestions for Authors

In this work, a model is presented for the calculation of the pressure drops of the concrete flow as a function of the pipe curvature. While the topic is interesting there are significant concerns that need to be addressed to enhance the clarity, depth, and rigor of the manuscript.

1. The meaning of the coefficients in Eq. 4, in particular of a, b, h and r must be explained more in the text.

2. Line 182 Cd is missing in Eq. 11 and 12.

3. The section mesh in Fig. 2 is different from Fig. 3. Which mesh was used?

4. The description of Fig. 3b and 3c is missing. Please add in the text.

5. Some numerical aspects are missing. Under which conditions is the convergent solution considered to be reached? Is Kε a model suitable for the simulated case? If possible add some references about this.

6. It would be better to add some explanation about the experimental results. How were they obtained?

7. Figs. 9, 10 and 11 would perhaps be more readable if R2 was reported in a separate tables as a function, respectively of r, Q and θ

8. In the tab.4 under the first column "Pump pipe inclination angle" is sufficient to indicate the values (0°, 45°, 90°...), without rewriting for each row "horizontal inclination angle 0°", "horizontal inclination angle 45°". Moreover sometimes it comes written "Pump pipe inclination angle" and other "horizontal inclination angle" this can be confusing. Choice one way and stick with that. Always in the Tab. 4 is not reported that the all the values under the columns of curvature radius are pressure loss and the units is missing.

9. Concrete flow is a very difficult mixture to simulate. In this type of work there is often a calibration phase that links the simulation with the real behavior of the material. In the introduction this aspect should also be mentioned by citing the paper DOI 10.1007/s40430-020-02464-6 and similar.

10. In the discussion the limitations of the model should be mentioned. There are different concrete consistency (slump) and these depend on mixing time, by the mixture, the aggregates etc.. I suppose that depending on the slump of the concrete the pressure losses are different. So it should be highlighted, also in the conclusions that the results obtained are valid for that slump.

Round 2

Reviewer 1 Report

Comments and Suggestions for Authors

Dear Authors,

I appreciate your effort and conducted corrections. I must emphasize that the title should be  “The One-Dimensional Flow Pressure Loss Correction Model Based on the Particle Flow through Concrete Bend”

Also, as I have mentioned before, Modi does not exist! It is Moody. He made a Chart, called Moody¢s Chart used to determine the coefficient of friction as a function of Reynolds number (Re=vD/n) and relative roughness. Everywhere in the paper should be changed from Modi to Moody! (Line 263, Also, for EDITORS, it is important that some correction is made in the paper „One-Dimensional Modeling of the Pressure Loss in Concrete Pumping and Experimental Verification“ by Xuan Zhao, Guoqiang Gao, Minshun Wan and Juchuan Dai, published in MDPI concerning this.

I have noticed following mistakes:

Line 193 and 194 Re, not Re!

 Line 417 and 418: make space between number and unit (3 corrections)

Author Response

请参阅附件。
